# GSK3β Inhibition Reduced Vascular Calcification in *Ins2^Akita/+^* Mice

**DOI:** 10.3390/ijms24065971

**Published:** 2023-03-22

**Authors:** Kristina I. Boström, Xiaojing Qiao, Yan Zhao, Xiuju Wu, Li Zhang, Jocelyn A. Ma, Jaden Ji, Xinjiang Cai, Yucheng Yao

**Affiliations:** 1Division of Cardiology, David Geffen School of Medicine at UCLA, Los Angeles, CA 90095-1679, USA; 2The Molecular Biology Institute at UCLA, Los Angeles, CA 90095-1570, USA

**Keywords:** vascular calcification, endothelial cells, diabetes mellitus, glycogen synthase kinase-3β inhibition

## Abstract

Endothelial–mesenchymal transition (EndMT) drives the endothelium to contribute to vascular calcification in diabetes mellitus. In our previous study, we showed that glycogen synthase kinase-3β (GSK3β) inhibition induces β-catenin and reduces mothers against DPP homolog 1 (SMAD1) to direct osteoblast-like cells toward endothelial lineage, thereby reducing vascular calcification in *Matrix Gla Protein (Mgp)* deficiency. Here, we report that GSK3β inhibition reduces vascular calcification in diabetic *Ins2^Akita/wt^* mice. Cell lineage tracing reveals that GSK3β inhibition redirects endothelial cell (EC)-derived osteoblast-like cells back to endothelial lineage in the diabetic endothelium of *Ins2^Akita/wt^* mice. We also find that the alterations in β-catenin and SMAD1 by GSK3β inhibition in the aortic endothelium of diabetic *Ins2^Akita/wt^* mice are similar to *Mgp^−/−^* mice. Together, our results suggest that GSK3β inhibition reduces vascular calcification in diabetic arteries through a similar mechanism to that in *Mgp^−/−^* mice.

## 1. Introduction

Vascular calcification is a common and severe complication of diabetes mellitus which affects more than 9% of the American population [1,2,3,4,5]. Previously considered to be a passive process of mineral precipitation, vascular calcification is now known as an active process that involves ectopic bone formation [1,2,3,4]. In this process, dysregulated systemic and local factors compel vascular cells to switch cell fates to osteogenic differentiation [1,2,3,4,5,6,7,8,9,10]. In diabetes mellitus elevated by hyperglycemia, bone morphogenetic protein (BMP) signals drive endothelial cells (ECs) to transdifferentiate into osteoblast-like cells causing arterial calcification [5,7,8,9,10]. Indeed, the role of the endothelium in vascular calcification is not limited to being a source of osteoinductive factors responding to hyperglycemia, oscillatory shear stress, or hyperlipidemia [9,11,12]. It also directly contributes to the calcifying process of osteoprogenitor cells [5,6,7,8,9,10,13,14,15]. Osteoblast-like cells with EC origin can be detected in calcified lesions of diabetic aortic tissues and atherosclerotic plaques [5,6,7,8,9,12]. Mechanistically, ECs are driven by endothelial-mesenchymal transitions (EndMTs) to gain plasticity and differentiate into osteoblast-like cells [5,7,8,9,10,14].

Previous studies have demonstrated mechanistic similarities between vascular calcification in Matrix Gla Protein (MGP) deficiency and diabetic models. These include increased BMP activity in the aortic endothelium of *Mgp^−/−^* mice, diabetic *Ins2^Akita/+^* mice, *db/db* mice, and human islet amyloid polypeptide transgenic rats [9]. The elevated BMP signaling stimulates the endothelium to contribute cells to the calcifying process in *Mgp^−/−^* and diabetic *Ins2^Akita/+^* aortas [5,7,8]. The increased BMP induced by high glucose in ECs drives ECs toward osteogenic differentiation through EndMTs [5,6,7,8,9]. The EC-derived osteoblast-like cells are detected in calcified lesions in diabetic aortas by lineage tracing, and the differentiation of these cells and calcification are limited by BMP inhibition [6,7,9]. 

The *Ins2^Akita/wt^* mouse results from the Akita mutation, which largely reduces mature insulin by disruption of the two disulfide bonds of A and B chains [16]. *Ins2^Akita/wt^* mice become spontaneously diabetic at 3–4 weeks of age and are recognized as a model for type I diabetes mellitus (DM1) [17]. Previous studies have shown that *Ins2^Akita/wt^* mice develop vascular calcification and provide not only a monogenic diabetic model but also a model of diabetic calcific vasculopathy [5]. 

Glycogen synthase kinase 3 (GSK3) is a serine/threonine kinase that is constitutively activated in unstimulated cells [18]. The activity of GSK3 is regulated by serine phosphorylation in response to extracellular signals [19]. GSK3 plays different roles in osteogenic and endothelial differentiation. GSK3 promotes osteogenic differentiation [20], and GSK3 deficiency disrupts the maturation of osteoblasts, resulting in the reduction of bone formation [21]. In contrast, GSK3 prevents endothelial differentiation, and inhibition of GSK3 promotes the differentiation, proliferation, and migration of ECs [22,23]. GSK3 has two isoforms, GSK3a and GSK3β. SB216763 is a small molecule compound that specifically inhibits the activity of the GSK3 isoforms in an ATP-competitive manner [24]. SB216763 has been commonly used to probe the function of GSK3 inhibition [18]. 

In a previous study, our results suggested that the aortic osteoblast-like cells were redirected back to endothelial differentiation by the SB216763 treatment in *Mgp^−/−^* mice [25]. We also showed that osteoblast-like cells with EC origin contributed to aortic calcification in *Ins2^Akita/+^* mice [7]. Here, we hypothesize that GSK3β inhibition ameliorates vascular calcification in diabetic *Ins2^Akita/+^* mice. 

## 2. Results

### 2.1. GSK3β Inhibition Reduced Aortic Calcification in Ins2^Akita/+^ Mice

To determine whether GSK3β inhibition reduces aortic calcification in *Ins2^Akita/+^* mice, we treated *Ins2^Akita/+^* mice with SB216763 (5 µg/g daily) or saline control at 36 weeks of age for 4 weeks. Alizarin red staining showed a robust decrease of aortic calcification in the SB216763-treated *Ins2^Akita/+^* mice (Figure 1a,b). The quantification of total aortic calcium confirmed the reduction of calcium in the mice (Figure 1c). Immunoblotting of whole aortic tissues showed that SB216763 reduced the expression of osteogenic markers, osterix and osteocalcin (Figure 2a,b). Together, the results suggested that SB216763 reduced the calcification in diabetic *Ins2^Akita/+^* mice.

### 2.2. GSK3β Deletion Limited Aortic Calcification in Ins2^Akita/+^ Mice

Previously, we performed lineage-tracing using *Col1α1^CreERT2^* mice and identified osteoblast-like cells in *Mgp^−/−^* aortic tissues [25]. We showed that osteoblast-specific deletion of GSK3β reduced aortic calcification in *Mgp^−/−^* mice [25]. To determine if osteoblast-specific deletion of GSK3β ameliorates vascular calcification in diabetes mellitus, we treated mice at 34 weeks of age with tamoxifen (75 mg/kg, daily) for 5 days to delete GSK3β in osteoblast-like cells as previously described [25]. At 40 weeks of age, we examined the aortic tissues. Alizarin red staining showed reduced calcification in the *Col1a1^CreERT2^ GSK3β^flox/flox^Ins2^Akita/+^* mice (Figure 3a,b). Total aortic calcium was also significantly decreased in the mice with GSK3β deletion (Figure 3c). Immunoblotting of whole aortic tissues revealed the reduction of osteogenic markers in *Col1a1^CreERT2^ GSK3β^flox/flox^Ins2^Akita/+^* mice after GSK3β deletion (Figure 4a,b). The results suggested that osteoblast-specific deletion of GSK3β reduced the calcification in diabetic *Ins2^Akita/+^* mice.

### 2.3. GSK3β Inhibition Redirected Osteoblast-like Cells toward Endothelial Differentiation in Ins2^Akita/+^ Mice

To determine if GSK3β inhibition directed EC-derived osteoblast-like cells in *Ins2^Akita/+^* mice to revert endothelial differentiation, we generated *VE-cadherin^creERT2^Rosa^tdTomato^Ins2^Akita/+^* mice. At 18 weeks of age, we treated the mice with tamoxifen (75 mg/kg, daily) for 5 days to label the aortic ECs as previously described [25]. At 20, 30, and 40 weeks of age, we isolated tdTomato-positive aortic cells and examined the endothelial and osteogenic markers (Figure 5a). Real-time PCR showed a decrease of endothelial markers with an increase of osteogenic markers in the tdTomato-positive cells of *Ins2^Akita/+^* mice (Figure 5b). We treated the mice with SB216763 (5 µg/g daily) at 36 weeks of age for 4 weeks and isolated tdTomato-positive cells (Figure 5c). The results showed that SB216763 prevented the decrease of endothelial markers and inhibited the increase of osteogenic markers (Figure 5d), suggesting that GSK3β inhibition directed EC-derived osteoblast-like cells back to endothelial differentiation in *Ins2^Akita/+^* mice. 

Previous studies showed that in *Mgp^−/−^* mice, SB216763 increased β-catenin, thereby suppressing SMAD1 and osteoblastic fate but stimulating β-catenin and endothelial differentiation [25]. Here, we examined β-catenin and SMAD1 in tdTomato-positive aortic cells of *VE-cadherin^creERT2^Rosa^tdTomato^Ins2^Akita/+^* mice treated with SB216763. Immunoblotting showed increased β-catenin but decreased SMAD1 in the tdTomato-positive cells of the SB216763-treated group (Figure 6a,b), suggesting that GSK3β inhibition guided EC-derived osteoblast-like cells back to endothelial differentiation in *Ins2^Akita/+^* mice, similar to the findings in *Mgp^−/−^* mice.

## 3. Discussion

This study provides evidence that GSK3β inhibition reduces vascular calcification in diabetes mellitus. The role of GSK3 in diabetes mellitus has been well investigated in recent studies. GSK3 activity was found to regulate insulin sensitivity, which directly affects glycogen synthesis and glucose metabolism [26]. Several signaling pathways are involved in these processes, such as the phosphatidylinositol 3-kinase (PI3K)/protein kinase B (AKT) signaling pathway [26]. Interestingly, inhibition of GSK3 improves the activity of glycogen synthase and glucose uptake, pointing to GSK3 inhibition as a potential antidiabetic strategy [27]. However, the activation of GSK3 in diabetic vascular calcification has never been reported, although GSK3 has been shown to promote osteogenic differentiation and bone formation [20,21]. GSK3β is one of the GSK3 isoforms. Here, we find that limiting GSK3β reduces the calcification in diabetic mice and reveals that the GSK3β inhibitor SB216763 redirects osteoblast-like cells to endothelial differentiation, similar to our previous observations in *Mgp^−/−^* mice [25]. These findings might bring more attention to GSK3 inhibition as a strategy to limit diabetes and calcification. 

EndMTs have been observed to contribute to vascular calcification in diabetes mellitus. Previous studies demonstrated that ECs lose their normal morphology but express mesenchymal stem cell markers to migrate through a degraded internal elastic lamina into the arterial media and contribute to calcification [6,7,10]. The studies showed that excess BMP activity induces a number of serine proteases, such as elastases and kallikreins, to activate Sry-box 2(Sox2) expression in ECs and trigger EndMTs toward osteogenic differentiation [28]. A recent study constructed a systematic screen to explore the possibility of re-directing osteoblast-like cells in vascular calcification back to endothelial differentiation. The GSK3β inhibitor SB216763 was identified to have this capacity and decreased vascular calcification in *Mgp^−/−^* mice [25]. In this study, the GSK3β inhibitor SB216763 also reduced vascular calcification in diabetic *Ins2^Akita/+^* mice. Our results suggest that GSK3β inhibition prevents EndMTs and reduces calcification in diabetes. 

## 4. Methods

### 4.1. Animals 

*Ins2^Akita/+^* (C57BL/6-*Ins2^Akita^*/J), *GSK3β^flox/flox^* (B6.129(Cg)-Gsk3b^tm2Jrw^/J), *Col1α1^CreERT2^* (B6.Cg-Tg(Col1α1-cre/ERT2)1Crm/J), and B6.Cg-*Rosa^tdTomato^*Gt(ROSA)^26Sortm9(CAG−tdTomato)Hze^/J mice were obtained from the Jackson Laboratory. The *VE-cadherin^cre/ERT2^* mouse was obtained as a gift from Dr. Ralf Adams. Genotypes were confirmed by PCR [29], and experiments were performed with generations F4–F6. Littermates were used as wild-type controls. All mice were fed a standard chow diet (Diet 8604, Harlan Teklad Laboratory, Indianapolis, Indiana, United States). The studies were reviewed and approved by the Institutional Review Board and conducted in accordance with the animal care guideline set by the University of California, Los Angeles. The investigation conformed to the National Research Council, *Guide for the Care and Use of Laboratory Animals, Eighth Edition* (Washington, DC, USA: The National Academies Press, 2011). We bred *Col1a1^CreERT2^GSK3β^flox/flox^* mice with *Ins2^Akita/+^* mice to create *Col1a1^CreERT2^ GSK3β^flox/flox^Ins2^Akita/+^* mice. SB216763 (Sigma-Aldrich, S3442) was injected via tail vein or retro-orbital injection (5 µg/g, daily) as in previous studies [30]. Injections of the *Ins2^Akita/+^* mice and wild-type mice started at 36 weeks of age and continued for 4 weeks. 75 mg/kg of tamoxifen (Sigma-Aldrich, T5648) was injected daily for 5 days. 

### 4.2. RNA Analysis

Real-time PCR analysis was performed as previously described [25]. Glyceraldehyde 3-phosphate dehydrogenase (GAPDH) was used as a control gene. Primers and probes for mouse VE-cadherin (Mm00486938_m1), osterix (Mm00504574_m1), osteocalcin (Mm03413826_mH), Flk1 (Mm01222421_m1), and von Willebrand factor (Mm00550376_m1) were obtained from Applied Biosystems as part of Taqman^®^ Gene Expression Assays.

### 4.3. Pre-Sorting of tdTomato Positive Cells

The pre-sorting of aortic tdTomato positive cells was performed as previously described [25]. The aortas were perfused briefly with dispase and enzymatically dispersed. Then, the aortas were dissected into small pieces and incubated for 45 min prior to cell isolation, fixation, staining, and FACS analysis.

### 4.4. Immunoblotting

Immunoblotting was performed as previously described [17]. Equal amounts of cellular protein or tissue lysates were used. These include SMAD1 (Cell Signaling Technology, 9743), β-catenin (R&D system, AF1329. Minneapolis, MN. USA), osterix (Santa Cruz Biotechnology, sc-22536. Dallas, TX, USA), Flk1 and VE-cadherin (all from BD Bioscience, 55,307 and 562,242, San Jose, CA, USA), and vWF (Dako, A0082, Santa Clara, CA, USA). β-Actin (Sigma-Aldrich, A2228. Saint Louis, MO, USA) was used as a loading control.

### 4.5. Quantification of Aortic Calcium

Total aortic calcium was measured using a calcium assay kit (Bioassay) as previously described [25]. 

### 4.6. Alizarin Red Staining

Slides were stained with Alizarin red solution (2% Alizarin red in distilled water) for 2 min. Then, excess solution was removed. The sections were dehydrated in acetone, followed by acetone-xylene (1:1) solution. After that, the sections were cleared by xylene and mounted with mounting medium.

### 4.7. Lesion Quantification

The mice were euthanized, and then perfusion fixed with 10% buffered formalin via the left ventricle for 4 min. The proximal aorta was excised. The specimen was embedded in OCT (Tissue-Tek, Fisher Scientific, Waltham, MA, USA), frozen on dry ice, and stored at −80 °C until sectioning. Serial cryosections were prepared. From then on, every fifth 10-μm section was collected on poly-D-lysine–coated slides. Sections were stained with hematoxylin and Alizarin red. Slides were examined by light microscopy, and the lesion area was quantified with computer-assisted image analysis (Image-Pro Plus, Media Cybernetics, Bethesda, MD, USA). 

### 4.8. Statistical Analysis

Data were analyzed for statistical significance by ANOVA with post hoc Tukey’s analysis. The analyses were performed using GraphPad Instat^®^, version 3.0 (GraphPad Software). Data represent mean ± SD. *p* < 0.05 was considered significant, and experiments were performed a minimum of three times.

## 5. Conclusions

Together, our results show the importance of GSK3 inhibition in diabetic vascular calcification and provide new information for developing new treatment strategies. 

## Figures and Tables

**Figure 1 ijms-24-05971-f001:**
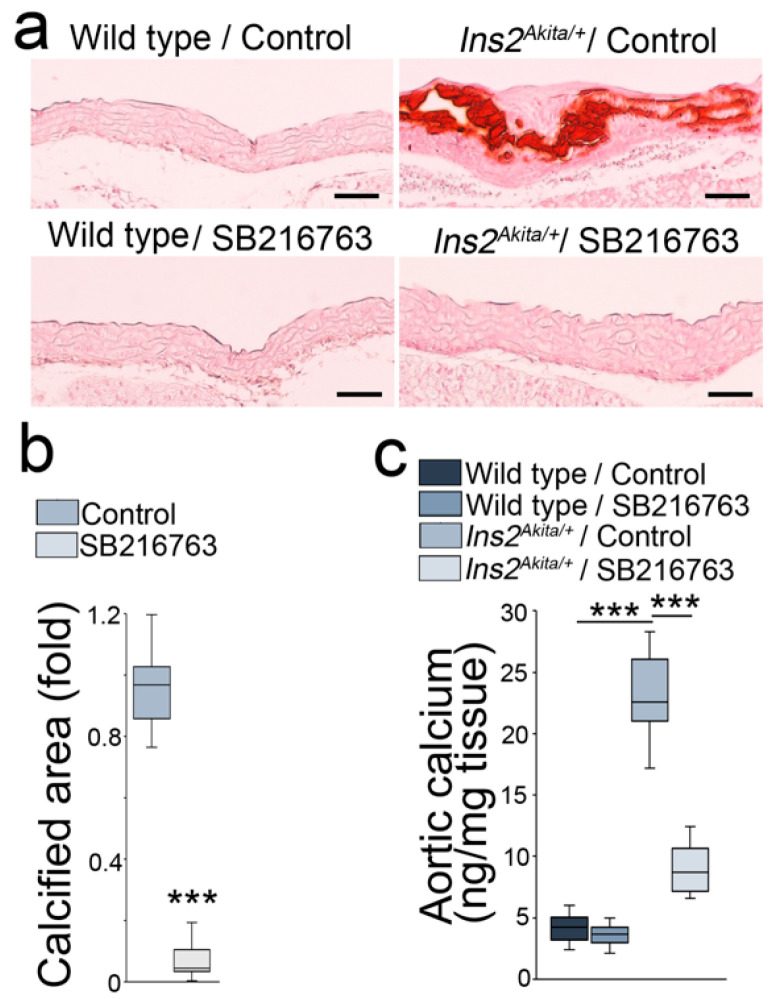
GSK3 inhibition reduces aortic calcification of *Ins2^Akita/+^* mice. (**a**) Alizarin red staining of aortas of *Ins2^Akita/+^* mice and wild-type mice treated with or without SB216763 (*n* = 8). (**b**,**c**) Calcified aortic area and total aortic calcium of *Ins2^Akita/+^* mice treated with or without SB216763 (*n* = 10). Scale bar, 50 µm. (**b**) was analyzed for statistical significance by unpaired 2-tailed Student’s *t*-test. (**c**) was analyzed for statistical significance by ANOVA with post hoc Tukey’s analysis. *** *p* < 0.001.

**Figure 2 ijms-24-05971-f002:**
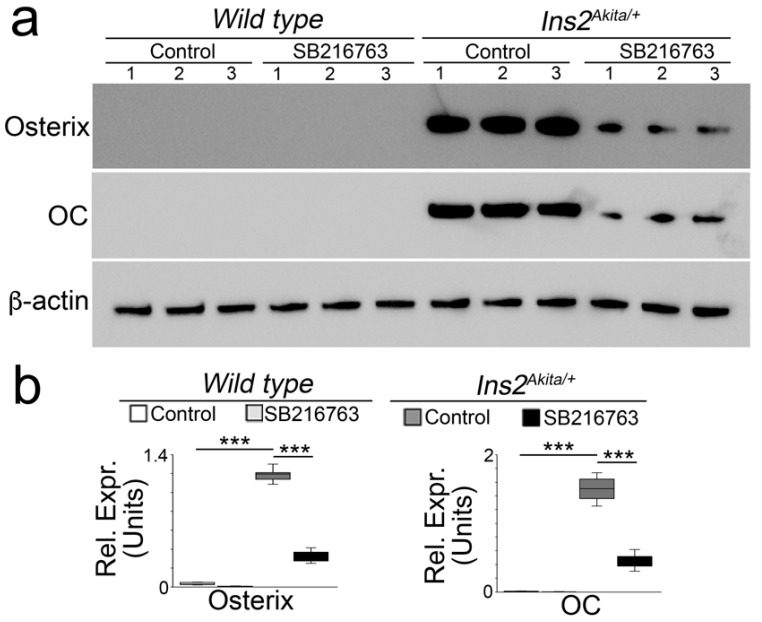
Immunoblotting (**a**) with densitometry analysis, and (**b**) of aortic tissues of *Ins2^Akita/+^* mice and wild-type mice treated with or without SB216763 (*n* = 3). The numbers represent each sample. OSX, osterix; OC, osteocalcin. Densitometry was analyzed for statistical significance by ANOVA with post hoc Tukey’s analysis. *** *p* < 0.001.

**Figure 3 ijms-24-05971-f003:**
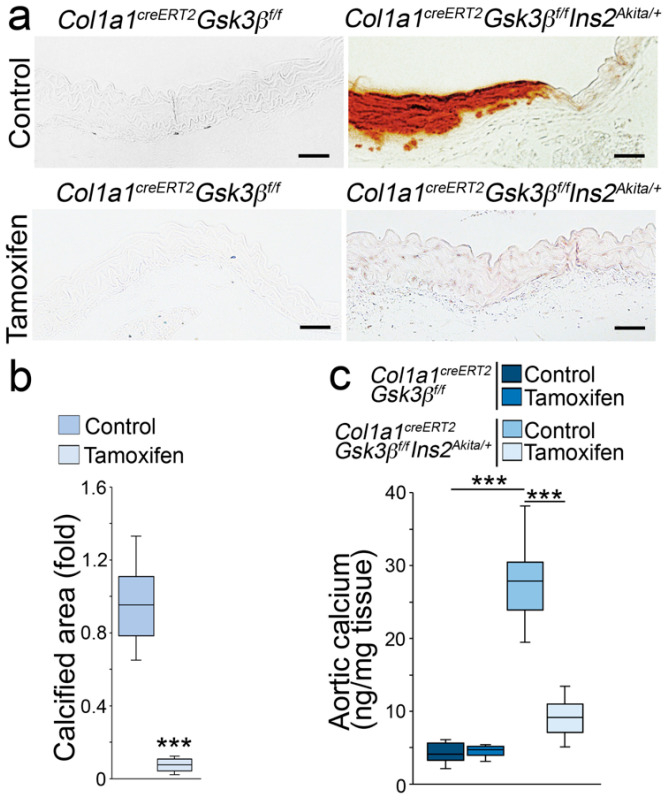
Deletion of GSK3β decreases calcification in *Ins2^Akita/+^* aortas. (**a**) Alizarin red staining of the aortas of Col1a1^creERT2^Gsk3β^flox/flox^*Ins2^Akita/+^* and Col1a1^creERT2^Gsk3β^flox/flox^ mice after tamoxifen treatment (*n* = 8). (**b**,**c**) Calcified aortic area and total aortic calcium of Cola1^creERT2^ Gsk3β^flox/flox^*Ins2^Akita/+^* and Cola1^creERT2^Gsk3β^flox/flox^ mice after tamoxifen treatment (*n* = 8). (**b**) was analyzed for statistical significance by unpaired 2-tailed Student’s *t*-test. (**c**) was analyzed for statistical significance by ANOVA with post hoc Tukey’s analysis. *** *p* < 0.001. Scale bar, 50 µm.

**Figure 4 ijms-24-05971-f004:**
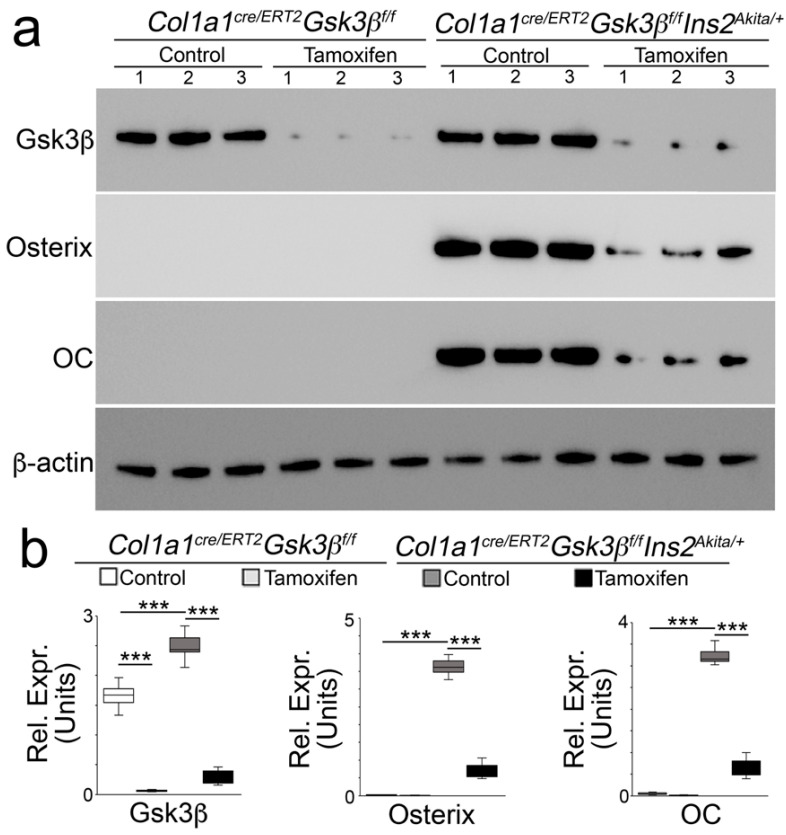
Immunoblotting (**a**) with densitometry analysis, and (**b**) of aortic tissues of Col1a1^creERT2^Gsk3β^flox/flox^*Ins2^Akita/+^* and Col1a1^creERT2^Gsk3β^flox/flox^ mice after tamoxifen treatment (*n* = 3). The numbers represent each sample. Densitometry was analyzed for statistical significance by ANOVA with post hoc Tukey’s analysis. *** *p* < 0.001.

**Figure 5 ijms-24-05971-f005:**
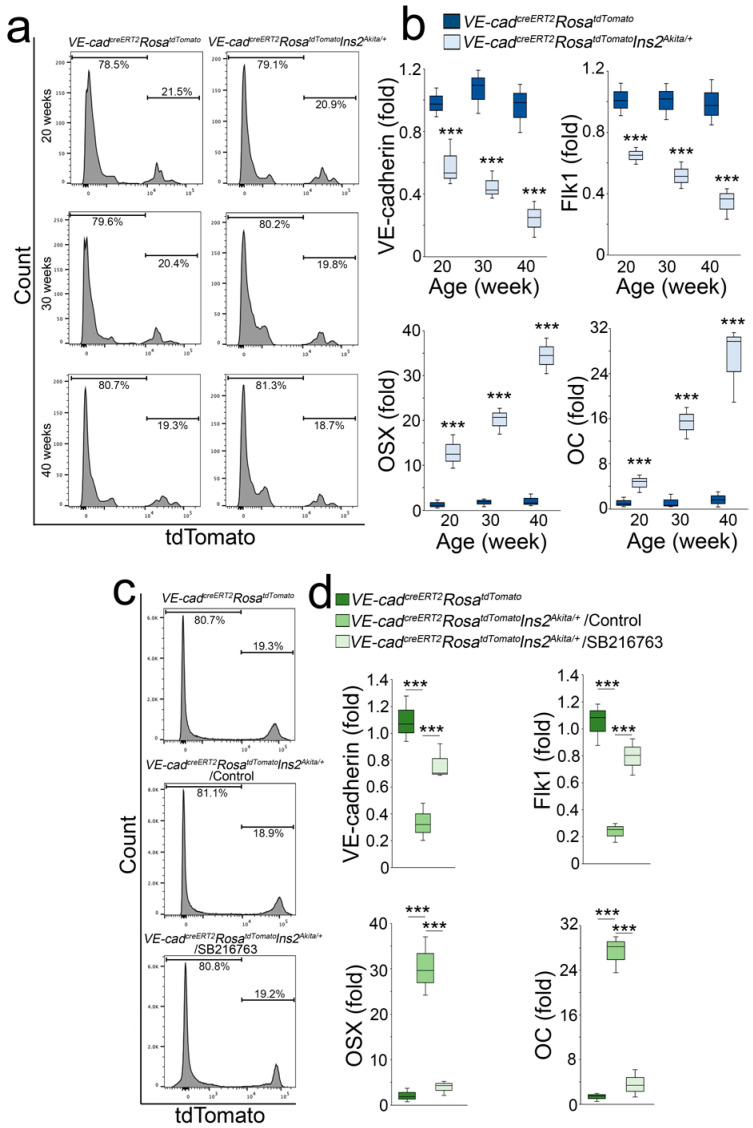
SB216763 reduces osteogenic markers and restores endothelial markers in EC-derived osteoblast-like cells. (**a**) FACS analysis of aortic tdTomato-positive cells of VE-cadherin^creERT2^Rosa^tdTomato^*Ins2^Akita/+^* and VE-cadherin^creERT2^Rosa^tdTomato^ mice at different ages. (**b**) Expression of VE-cadherin, Flk1, osterix (OSX), and osteocalcin (OC) in the tdTomato-positive cells isolated from VE-cadherin^creERT2^Rosa^tdTomato^*Ins2^Akita/+^* and VE-cadherin^creERT2^Rosa^tdTomato^ mice at different ages (*n* = 3). (**c**) FACS analysis of aortic tdTomato-positive cells of VE-cadherin^creERT2^Rosa^tdTomato^*Ins2^Akita/+^* and VE-cadherin^creERT2^Rosa^tdTomato^ mice treated with SB216763. (**d**) Expression of VE-cadherin, Flk1, OSX, and OC in the tdTomato-positive cells isolated from VE-cadherin^creERT2^Rosa^tdTomato^*Ins2^Akita/+^* and VE-cadherin^creERT2^Rosa^tdTomato^ mice treated with SB216763 (*n* = 3). (**b**) was analyzed for statistical significance by unpaired 2-tailed Student’s *t*-test. (**d**) was analyzed for statistical significance by ANOVA with post hoc Tukey’s analysis. *** *p* < 0.0001.

**Figure 6 ijms-24-05971-f006:**
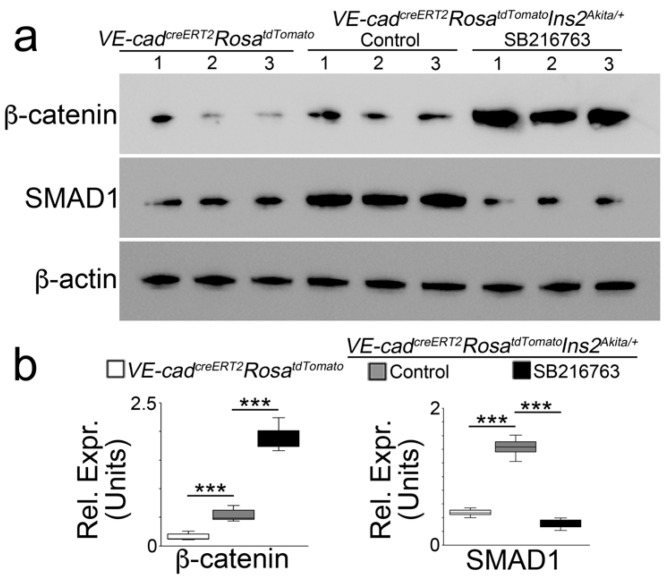
Immunoblotting (**a**) with densitometry analysis, and (**b**) of the tdTomato-positive cells isolated from VE-cadherin^creERT2^Rosa^tdTomato^*Ins2^Akita/+^* and VE-cadherin^creERT2^Rosa^tdTomato^ mice treated with or without SB216763 (*n* = 3). The numbers represent each sample. Densitometry was analyzed for statistical significance by ANOVA with post hoc Tukey’s analysis. *** *p* < 0.001.

## Data Availability

No new data were created.

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
