# Peer review of "GSK3β Inhibition Reduced Vascular Calcification in Ins2Akita/+ Mice"

_ijms, 2023, doi:10.3390/ijms24065971_

Round 1
Author Response
Reviewer #1
- Figure 1D, 2D and 3C reflect only one experience. It would be preferable to make a new figure including a graph with statistics with your three independent experiments (as indicated in your legends n=3). All original images for blots (n=3) must be filed (indicating molecular weight). Would it be possible to improve the quality of the SMAD1 blot?
We repeated the immunoblotting for the three repeats as requested. Densitometry was performed for each blot. We created new Figures 2, 4 and 6.
- Figure 3A: Could you include FACS analysis and selection gate for tdTomato positive aortic cells at 20, 30 and 40 weeks?
FACS analysis was added to the figure (Figure 5a).
- Figure 3A: Could you include FACS analysis and selection gate for tdTomato positive aortic cells for the three mouse models?
FACS analysis was added to the figure (Figure 5c).

Reviewer 2 Report
In this study, Bostrom et al. explored the effects of GSK3beta deficiency on diabetes mellitus type I mouse models. The methods are generally appropriate. The results are clear, support the conclusions, and are interesting. On the other hand, the manuscript is poorly written.
Introduction:
- The introduction does not provide sufficient background on vascular calcification.
- It would be helpful to briefly introduce the role of GSK3beta to provide some context. Of the three subjects from the title (GSK3beta, Vascular Calcification, Ins2Akita/+ 2 Mice), only the mouse model is sufficiently explained. If you have a character limit, I would suggest condensing the BMP signaling some.
- Also, abbreviations (MGP) must be explained when they appear for the first time in the text.
- It would be appropriate to add your reference/s in line 41.
- I would advise rewriting the introduction to provide a better flow (arrangement) of information.
Methods:
- This section is missing information. There is currently not enough detail on the methodology and characterization of some of the techniques used. It would be advisable to describe the methods used to avoid self-citation and overuse of "as previously described".
- I would advise adding the sequences of the primers used.
- There is no mention of Alizarin red staining in this section.
- There is no mention of tamoxifen in this section.
- Authors should specify which ANOVA they used.
Results:
- It is not appropriate to describe results from your past studies in this section. Only current results should be here. It is difficult to come by what you did in this study otherwise.
- Also, some methods are described here, but not in the methods section. Sentences like the ones in lines 108-110 should be relocated.
- In Figure 1 we can see that Wild type mice were also treated with SB216763. If that is so, the four separate groups need to be mentioned in the methods section.
- How was the calcified area calculated? This should be explained in the methods section.
- What do the n numbers in the figure legends refer to? How many WT and KO mice were assessed for each experiment?
- No molecular weight indicators are provided with any of the original blots.
- Please change the capital D in Figure 1 to match the other letter indicators of panels. Also, the figure is a bit busy, but I find it justifiable considering the amount of information presented.
- In Figure 2, the letter indicators should be aligned properly.
- Lines such as 142-144 would be more appropriate in the discussion.
Discussion:
- There is mention of GSK3, GSK-3 and GSK3beta in this section. If there is any difference between them, this should be noted. If not, they should be matched to avoid confusion.
Author Response
Reviewer #2
Introduction:
- The introduction does not provide sufficient background on vascular calcification.
We added background on vascular calcification to the introduction.
- It would be helpful to briefly introduce the role of GSK3beta to provide some context. Of the three subjects from the title (GSK3beta, Vascular Calcification, Ins2Akita/+ 2 Mice), only the mouse model is sufficiently explained. If you have a character limit, I would suggest condensing the BMP signaling some.
An introduction to GSK3b was added.
- Also, abbreviations (MGP) must be explained when they appear for the first time in the text.
The text has been updated.
- It would be appropriate to add your reference/s in line 41.
The reference was added.
- I would advise rewriting the introduction to provide a better flow (arrangement) of information.
The introduction was updated.
Methods:
- This section is missing information. There is currently not enough detail on the methodology and characterization of some of the techniques used. It would be advisable to describe the methods used to avoid self-citation and overuse of "as previously described".
This section has been updated.
- I would advise adding the sequences of the primers used.
The ID# of each Taqman assay has been added.
- There is no mention of Alizarin red staining in this section.
Alizarin red staining has been added.
- There is no mention of tamoxifen in this section.
Tamoxifen injection has been mentioned in the method.
- Authors should specify which ANOVA they used.
Statistical analyses were added to each figure legend.
Results:
- It is not appropriate to describe results from your past studies in this section. Only current results should be here. It is difficult to come by what you did in this study otherwise.
The description of the previous study has been relocated to the introduction.
- Also, some methods are described here, but not in the methods section. Sentences like the ones in lines 108-110 should be relocated.
It has been relocated to the method section.
- In Figure 1 we can see that Wild type mice were also treated with SB216763. If that is so, the four separate groups need to be mentioned in the methods section.
This has been corrected.
- How was the calcified area calculated? This should be explained in the methods section.
The method for calculating the calcified area has been added.
- What do the n numbers in the figure legends refer to? How many WT and KO mice were assessed for each experiment?
The n number refers to the number of animals in each experimental group presented in the figure.
- No molecular weight indicators are provided with any of the original blots.
Indicators have been added.
- Please change the capital D in Figure 1 to match the other letter indicators of panels. Also, the figure is a bit busy, but I find it justifiable considering the amount of information presented.
The figure has been updated, now as Figure 2.
- In Figure 2, the letter indicators should be aligned properly.
The figure has been updated, now as Figure 3.
- Lines such as 142-144 would be more appropriate in the discussion.
This sentence was used to link this study to the previous study, which would make it easier to interpret the results. We would like to keep it as is.
Discussion:
- There is mention of GSK3, GSK-3 and GSK3beta in this section. If there is any difference between them, this should be noted. If not, they should be matched to avoid confusion.
The discussion has been updated.
Round 2
Author Response
The Figures have been updated.
Reviewer 2 Report
Thank you for your revised manuscript.
I have found that all my major concerns have been addressed and therefore recommend this manuscript for publication in its present form.
Author Response
Thank you.